# Gut Symbiotic Microbial Communities in the IUCN Critically Endangered *Pinna nobilis* Suffering from Mass Mortalities, Revealed by 16S rRNA Amplicon NGS

**DOI:** 10.3390/pathogens9121002

**Published:** 2020-11-29

**Authors:** Athanasios Lattos, Ioannis A. Giantsis, Dimitrios Karagiannis, John A. Theodorou, Basile Michaelidis

**Affiliations:** 1Laboratory of Animal Physiology, Department of Zoology, Faculty of Science, School of Biology, Aristotle University of Thessaloniki, 54124 Thessaloniki, Greece; lattosad@bio.auth.gr (A.L.); michaeli@bio.auth.gr (B.M.); 2Department of Animal Science, Faculty of Agricultural Sciences, University of Western Macedonia, 53100 Florina, Greece; 3National Reference Laboratory for Mollusc Diseases, Ministry of Rural Development and Food, 54627 Thessaloniki, Greece; vetaquapath@gmail.com; 4Department of Animal Production Fisheries & Aquaculture, University of Patras, 30200 Mesolonghi, Greece; jtheo@upatras.gr

**Keywords:** *Pinna nobilis*, endangered species, microbiome, 16s-rRNA, NGS, metagenomics, *Vibrio* spp., *Mycoplasma* spp., *H. pinnae*, *Mycobacterium* sp.

## Abstract

Mass mortality events due to disease outbreaks have recently affected almost every healthy population of fan mussel, *Pinna nobilis* in Mediterranean Sea. The devastating mortality of the species has turned the interest of the research towards the causes of these events. After the haplosporidan infestation and the infection by *Mycobacterium* sp., new emerging pathogens have arisen based on the latest research. In the present study, a metagenomic approach of 16S rRNA next generation sequencing (NGS) was applied in order to assess the bacterial diversity within the digestive gland of diseased individuals as well as to carry out geographical correlations among the biodiversity of microbiome in the endangered species *Pinna nobilis*. The specimens originated from the mortalities occurred in 2019 in the region of Greece. Together with other bacterial genera, the results confirmed the presence of *Vibrio* spp., assuming synergistic effects in the mortality events of the species. Alongside with the presence of *Vibrio* spp., numerous bacterial genera were detected as well, including *Aliivibrio* spp., *Photobacterium* spp., *Pseudoalteromonas* spp., *Psychrilyobacter* spp. and *Mycoplasma* spp. Bacteria of the genus *Mycoplasma* were in high abundance particularly in the sample originated from Limnos island representing the first time recorded in *Pinna nobilis*. In conclusion, apart from exclusively the Haplosporidan and the Mycobacterium parasites, the presence of potentially pathogenic bacterial taxa detected, such as *Vibrio* spp., *Photobactrium* spp. and *Alivibrio* spp. lead us to assume that mortality events in the endangered Fan mussel, *Pinna nobilis,* may be attributed to synergistic effects of more pathogens.

## 1. Introduction

Microbial communities constitute the subject of numerous studies for many years, which have been mostly relied however, on culture dependent methods for detection, identification and characterization of virulence traits of various pathogens [1]. With the recent advances in molecular biology and bioinformatics, the theory of Holobionts arose to describe the interactions between the host and its symbiotic organisms as one entirety [2]. Bacterial diversity in bivalves is highly influenced by their filter feeding behavior with the ingestion of bacteria as well as by stressful conditions in their environment [3,4]. Research on the microbiota of aquatic animals has indicated its important role in many key functions such as immune responses-disease resistance, proper assimilation of nutrients through good intestine function and stimulation of biological processes [5,6]. This relationship is reciprocal for the microbiome and the host, due to the provision of a stable and nutritious environment provided by the host [7,8]. On the other hand, global warming because of climate change may result to lower immune responses and physiological performance in aquatic organisms [9]. As a result of the decline in immune responses, opportunistic pathogens, that may already belong to microbiome of the host, take advantage of that opportunity and multiply, creating colonies uncontrollably, to the detriment of the host [10,11].

Modern molecular tools, particularly Next Generation Sequencing (NGS) of the 16s rRNA gene, contributed to a rapid increase in the knowledge for microbial ecology. Symbiotic microbiota in invertebrates are of high importance as health indicators in ecosystems [12]. Using NGS, the interactions between a host and its microbiome, as well as the effects of environmental variables on these interactions can be also evaluated [4]. Considering that common microbiota, especially in the gut, are known to promote resistance against potential opportunist pathogens [13], the study and the comparison of the microbiota of mortality events with the microbiota of healthy individuals is of great interest. However, there are only few studies related with the microbiome within invertebrates and even fewer are associated with mortality events. In 2013, Bayer et al. [14], investigated the microbial assemblage of an ecosystem engineer gorgonian coral species of Mediteranean Sea, *Eunicella cavolini,* in different depths of a French Mediteranean coast. As a result, no distinct difference emerged in the structure of the microbiome of the corals of different water levels. However, bacteria from the genus *Endozoicomonas* were in abundance, which is frequently found to have association with several marine invertebrate species. In the context of climate change, Blanquer et al. [15], investigated the possible change in the diversity of the microbiome during the early detection of a disease in the sponge species *Ircinia fasciculata,* comparing microbiome of healthy individuals with microbiome of diseased individuals. The results showed a small shift, particularly in *Gammaproteobacteria* and *Acidobacteria* concentrations, while *Deltaproteobacteria* declined in diseased tissues versus healthy ones. Additionally, they observed an increase in bacterial diversity significantly in diseased tissues.

Concerning the microbiome of bivalves, Meisterhans et al. [16], studied the microbiota in each organ of the Manila clam, *Ruditapes philippinarum* and they tried to correlate the microbiome of each organ separately and also comparing with the environmental conditions of differentiated habitats. Another aim of this study was to evaluate the responses of *Ruditapes philippinarum* in a different habitat quality. Whereas there were differences in the microbiota from each organ, this study revealed no correlation with the particularities of environmental conditions of different habitats. Moreover, reciprocal transplantation experiments indicated that only in the degradation of the quality of the habitat had negative effects on the welfare and on microbiome structure, but not inversely. On the other hand, Vezzulli et al. [4], revealed that *Crassostrea gigas* is more susceptible in infections than the Mediterranean mussel *Mytilus galloprovinciallis*. Specifically, by using 16s rDNA gene based bacterial microbiota profile they showed that *C. gigas* is a better host for the growth of opportunist bacteria *Vibrio* spp. than *M. galloprovincialis*. Additionally, they reported that the microbiome of the Mediterranean mussel acts as an inhibitor for the growth of those bacteria. In 2018, Li et al. [17], investigated the impacts of diet and rise of temperature on the gut microbiota of the *Mytilus coruscus.* Microbial community in their project was investigated through the use of 16s rRNA gene sequencing. The results obtained from their study supported their hypothesis that increasing the temperature may react with the microbial diversity of the gut and favor the proliferation of opportunistic bacteria. No significant difference was observed in the gut microbiota with the use of microalgal diets. Later in 2019, Li et al. [18], investigated the microbial diversity in the hemolymph of the mussel *Mytilus coruscus* with the presence of *Vibrio cyclitrophicus* in different temperatures determined by Illumina Hiseq sequencing of 16s rRNA gene. Specimens in higher temperature (27 °C) suffered from greater mortality losses with the presence of *Vibrio cyclitrophicus* in the ninth day that was associated also with the decline in abundance in the bacterial diversity in the hemolymph. *Vibrio* presence promoted the proliferation of other opportunist bacteria at the low temperature (21 °C). They finally indicated that temperature is the key factor in shaping microbial communities. Milan et al. [19], investigated the microbial diversity of a mass mortality event occurred in the bivalve *Chamelea gallina* in association with the gene expression profile. Overexpression of several genes associated with xenobiotic metabolism and down regulation of genes associated with immune and stress responses were observed. In conclusion, they indicated that the great abundance of opportunistic bacteria is directly related with the low immune responses they observed from the gene profile. In 2019, Green et al. [20], simulated a mortality event episode in pacific oyster, *Crassostrea gigas*, raising the temperature rapidly (20 to 25 °C). They investigated the microbial diversity of this heat shock with and without the presence of antibiotics in the water. NGS of the 16s rRNA gene revealed the difference in microbial diversity between the examined temperatures, with a notable increase in *Vibrio* spp. abundance in parallel with the increase in temperature. The presence of the antibiotics showed the importance of the temperature in the proliferation of bacteria, showing differences up to 70% in mortality rates with the inhibitor factor i.e., the presence of antibiotics in bacterial growth.

*Pinna nobilis* is the largest marine bivalve in Mediterranean Sea and one of the largest worldwide, inhabiting depths between 0.5 m and 60 m, reaching maximum length up to 120 cm with lifespan that can overdraw 45 years [21,22,23]. Population diminishing of *P. nobilis* begun in 1980 as a consequence of anthropogenic activities. Illegal fishing for consumption or decorative purposes, natural habitat destruction and marine pollution were the causes of collapse of many populations in the Mediterranean Sea. Thus, the species went through a serious threat during the last 40 years on account of anthropogenic activities [24,25,26,27]. As a consequence of these threats, P. nobilis was included in a protection regime (Barcelona Convention and EU Habitats Directive) which led the species to a recovery of its populations, as already shown also by genetic analyses [25,26]. Unfortunately, recently, it was included in the IUCN Red List of threatened species, characterized as critically endangered [28]. *P. nobilis* populations suffer from mass mortality events, and risk of extinction in many of their habitats. This phenomenon has been investigated since the first mass mortality event occurred in the south-western Mediterranean coast of Spain, which caused mortalities in the local populations of fan mussel, whereas populations in the northern coasts seemed to be unaffected [29,30]. As causative agent for those mortalities is considered the protozoan parasite *Haplosporidium pinnae* which continued to devastate populations of fan mussel, *P. nobilis* in France, Italy, Tunisia, Turkey, Greece, reaching out one of the last safe shelters for the fan mussel in Croatia [22,31,32,33,34,35,36]. Strong inflammatory responses were observed in all infestation cases caused by *H. pinnae*, leading to infiltrations in the digestive gland. Nevertheless, *H. pinnae* is not the only pathogen recognized to infect fan mussels. In particular, bacteria belonging to the genus *Mycobacterium* were firstly recorded in fan mussel populations from Italy and Greece, also considered responsible for mortalities, as well as the opportunistic *Vibrio mediterranei* detected in stabled animals with association of stress observed in captivity [34,37,38,39].

It is still questionable, how the geographical patterns and the human activities of the sampling areas can affect the microbial diversity of the fan mussels. The fan mussel population of Thermaikos gulf was the last population collapsed, although there were no mortalities in the Spring and in early Summer months of 2019. In contrast, the populations in Lesvos island and in Limnos island, suffered from mass mortality events and their reproductive rate provides infected animals. Thus, the primary aims of this study were to investigate the variability and abundance of microbial communities in infected individuals of the endangered fan mussel, *P. nobilis*, applying 16s rRNA NGS technology and correlate them to each other. It is also crucial to investigate the bacterial communities and make correlations with the human activity in regional level as well as evaluate the consequences to the natural habitat of the endangered *P. nobilis*.

## 2. Materials and Methods

### Assessment of Gut Microbiome in P. nobilis from Greek Mass Mortality Events

Digestive gland tissues from *P. nobilis* specimens, originating from mass mortality events as described in Lattos et al. [34], (Table 1), were obtained to identify the microbial diversity in the gut of infected fan mussels. Three specimens originated from Thermaikos Gulf, Thessaloniki (Ther01, Ther02, Ther03), two specimens from Kalloni Gulf, Lesvos island (Kal10, Kal12) and specimen originated from Limnos island, North Aegean, were analyzed (Figure 1).

Among the analyzed samples as examined previously in Lattos et al. [34], Ther01 was infected only by *Mycobacterium* sp. while in Ther02 and Ther03 *H. pinnae* and *Mycobacterium* sp. were detected. Samples from Lesvos were infected either by both parasites, or solely by the *Mycobacterium* sp., whereas the sample Lim01 was the only representative from Limnos island, North Aegean, hosting both pathogens (Table 1). After the first investigation published in 2020, which indicated the presence of *H. pinnae* and *Mycobacterium* sp. [34], a monitoring study started in the end of 2019 and continues nowadays, indicating the presence of both parasites in every alive population existing in Greece. Hence, no healthy individual was included in the analysis.

Approximately 20 mg of the digestive gland were placed in a 1.5 mL sterile tube and proceeded for DNA extraction. QIAamp^®^DNA mini kit (Qiagen, Hilden, Germany) was used for isolation and purification of total microbial DNA, according to the protocol provided by the manufacturer company. The quality and the quantity of the extracted DNA was checked using a NanoDrop (Shimadzu, Kyoto, Japan). Library preparation was performed on the V4 region of 16s rRNA gene. Primers used for 16s rRNA gene amplification were those described by Caporaso et al. [40], and the sequencing procedure was implemented using Illumina Miseq technology. Sequence data were deposited to the NCBI database under the BioProject accession number PRJNA661537. Heatmaps demonstrating relative bacterial abundance were generated using Multiple Experiment Viewer (MEV) online version, whereas bacterial abundances where also depicted in bar graphs created in Microsoft excel.

## 3. Results

### 3.1. Gut Microbiota Phylum Analysis in Pinna nobilis

A total number of 1220 Operational Taxonomic Units (OTUs) were identified from the gastrointestinal gland of the fan mussel *P. nobilis* examined specimens. Seventeen different bacterial phyla were detected with 14 of them having >0.5% abundance in at least one sample analyzed (Figure 2). The bacteria of the phylum Proteobacteria, exhibited the greatest average abundance with 30.59% of the total microbiota identified, followed by the bacterial phyla Tenericutes and Fusobacteria with average abundances 13.18% and 12.16%, respectively. Moreover, the existence of Archaea, in particular of the phylum Crenarchaeota was also revealed. Concerning the specimens originated from Thermaikos Gulf, Thessaloniki, the dominant phyla in Ther02 and Ther03 were Fusobacteria and Proteobacteria, while in Ther01 the dominant phyla were Proteobacteria and Actinobacteria. Specifically, in the Ther01 Actinobacteria had the greatest abundance with 24.98% of the total microbiota and Proteobacteria had 22.99% respectively. In Ther03 Proteobacteria showed the greatest abundance of 41.74%, followed by Fusobacteria phylum with 17.34% of the total microbiome detected. In the sample Ther02, Fusobacteria was the dominant phylum (20.78%) with Proteobacteria slightly lower (20.38%). In both samples originated from Kalloni Gulf, Lesvos Island, the dominant bacterial phyla were Proteobacteria with abundance ratio 18.26% and 50.48%, for samples Kal10 and Kal12, respectively, followed by Fusobacteria, with 9.65% and 23.96%, respectively. Interestingly, in the sample Kal10, the phylum Firmicutes presented a differentiation in comparison with Kal12, with relative abundance ratio 8.72% compared to the 0.11% respectively. The gut microbiota in the sample of Limnos island followed a different pattern with dominant bacterial phyla Tenericutes and Proteobacteria, of 63.59% and 29.68% abundance, respectively.

### 3.2. Gut Microbiota Families Analysis in Pinna nobilis

The detection of 30 bacterial families was the result of the analysis of the OTUs with the highest relative abundance. Additionally, the analysis using the OTUs with relative abundance >0.5% and presence in at least one sample indicated 123 OTUs. A heatmap and a bar graph was created to demonstrate the relative abundance of the bacterial families in the gastrointestinal samples of Fan mussel (Figure 3). Mycoplasmataceae, Alteromonadaceae, Fusobacteriaceae, Vibrionaceae and Mycobacteriaceae were the five families with the most dominant average presence in the samples. In the sample coded Ther01, Mycobacteriaceae was the most dominant family with relative abundance at 14.24%. Comamonadaceae, Moraxellaceae, Mycoplasmataceae, Staphylococcaceae families followed with lower relative abundances i.e., 5.64%, 4.65%, 3.62% and 2.59% respectively. In Ther02, Fusobacteriaceae, Alteromonadaceae, Vibrionaceae, Mycoplasmataceae and Rhodobacteriaceae were the families with abundance rates 12.78%, 6.08%, 5.11%, 1.92%, 0.66% respectively. In the third sample from Thermaikos Gulf (Ther03) bacterial families with the highest abundance percentages were Alteromonadaceae, Fusobacteriaceae, Vibrionaceae, Mycoplasmataceae and Moritellaceae with 22.66%, 13.54%, 4.69%, 3.44% and 1.40% correspondingly. In Kal10, most abundant families were Fusobacteriaceae, Alteromonadaceae, Vibrionaceae, Erysipelotrichaceae and Clostridiales Family XII with rates 5.19%, 4.28%, 2.20%, 1.65 and 1.21% respectively. In the second sample from Lesvos island (Kal12) the most abundant microbiota familes detected were Alteromonadaceae, Fusobacteriaceae, Vibrionaceae, Moritellaceae and Colwelliaceae (22.33%, 14.35%, 8.45%, 1.64% and 1.56%). In the sample originated from Limnos island the diversity in the microbiota was scarce, with most abundant families to be Mycoplasmataceae, Vibrionaceae, Mycobacteriaceae, Moraxellaceae and Corynebacteriaceae with rates 63.59%, 25.38%, 0.07%, 0.02% and 0.02% respectively.

### 3.3. Gut Microbiota Genus Analysis in Pinna nobilis

A heatmap and a bar graph were constructed to demonstrate the most abundant bacterial genera within the Fan mussel, *P. nobilis* (Figure 4). Fourteen bacterial genera were selected among 1220 OTUs, having average abundance in the samples >1% and presence at least in one of the samples. In samples originated from Thermaikos Gulf the most abundant genera were *Mycobacterium sp. 1*, *Psychrilyobacter sp. 1* and *Vibrio sp. 2,* in Ther01, Ther02 and Ther03 respectively. *Vibrio sp. 1, Psychrilyobacter sp. 1 and Photobacterium sp.* were also representatives in the specimens of Thermaikos gulf, in higher abundance than the total average presence of bacteria in the samples (2.52%). In samples from Lesvos island, *Psychrilyobacter sp. 1, Vibrio sp. 1 and Vibrio sp. 2* were the most abundant genera in Kal10, while *Psychrilyobacter sp. 1, Vibrio sp. 1, Vibrio sp. 2, Photobacterium* sp., *Aliivibrio* sp. and *Pseudoalteromonas* sp. were the most abundant in Kal12. In the unique specimen from Limnos isl. the dominant genera were *Mycoplasma sp.1, Mycoplasma sp.2, Mycoplasma sp.3, Mycoplasma sp.4, Vibrio sp. 3, Vibrio sp. 4* and *Vibrio sp. 5*.

### 3.4. Blastn Search of Bacterial Genera

Blast search against NCBI database revealed 100% similarity of *Vibrio sp. 1* with two microbial species i.e., *Vibrio splendidus* and *Vibrio atlanticus*. Searches in the same database revealed 100% similarity of *Vibrio sp. 2* with many Vibrio species such as *V. toranzoniae, V. crassostreae, V. kanaloae, V. gallacicus, V. gigantis, V. chagasii, V. coralliirubri and V. pomeroyi.* Blastn search for *Vibrio sp. 3* revealed 100% similarity with *V. astriarenae*, 99.58% similarity with *V. agarivorans*, 99.58% similarity with *V. diabolicus* and the same again similarity with *V. mediterranei*. As a result of searching for *Vibrio sp*. 4, 100% similarity was revealed with *V. hippocampi*, alongside with lower similarities with a variety of *Vibrio* spp. such as *V. aestuarianus, V. alginolyticus, V. thalassae and V. mediterranei* with 98.74% similarity and *V. sinensis* with 98.73% similarity. Finally, Blast for *Vibrio sp. 5* demonstrated 98.5% similarities with *V. mediterranei* and *V. shilonii*. Mycoplasma sp. 3 was the only species in which Blast search revealed 84.57% similarities with *Candidatus Mycoplasma girerdii*, while other *Mycoplasma* spp. search leaded only in higher taxa classification.

## 4. Discussion

The present study demonstrated a great variability of symbiotic bacteria within the infected Fan mussel, *P. nobilis* specimens originating from mass mortality events in Greece. Among these bacterial species, several potential pathogens were determined. Generally, investigating the gut microbiome using 16s rRNA sequencing in bivalves, revealed a great bacterial diversity, many of which may have symbiotic or pathogenic role in the host. Carella et al. [39], confirmed the co-occurrence of other potential pathogens, alongside with *H. pinnae* and *Mycobacterium* sp. and in particular the existence of a new *Vibrio* sp. that infects *P. nobilis* populations and constitutes an opportunistic pathogen for this bivalve. *V. mediterranei* has been proved to be an opportunistic pathogen even for stabled individuals of *P. nobilis* and has been associated with mortalities in captured conditions [38]. As a result, in this study five different OTUs, genetically related with *Vibrio* sp. were confirmed in the specimens of all Greek regions investigated. Blastn search against NCBI database revealed high percentage similarities with many *Vibrio* species, which actually constitute pathogenic organisms in many cases for bivalves and are in relation with many mass mortality events in all life stages of marine bivalves.

The aquatic environment harbors a great abundance of bacteria. Bivalves, with their efficient filter feeding mechanism, may ingest a plethora of microorganisms mostly depending on the habitat. While most of the bacteria in the marine environment are not pathogenic to bivalves, unless their presence is characterized by a large number of colonies, some can be pathogenic or can utilize bivalves as an intermediate host in order to infect and become pathogenic to the final host. Within this diversity of microorganisms in the marine environment, the genus *Vibrio* seems to play a key role in the pathogenicity in marine bivalves. The genus *Vibrio* comprises more than 130 bacterial species which can be grouped in 17 monophyletic clades [41,42]. The vast majority of bacterial diseases of the bivalves have as causative agent the *Vibrio* spp. [43]. *V. aestuarianus* is an important pathogen responsible for massive mortalities in oyster, *Crassostrea gigas* in Europe and responsible for a greatly negative impact in farms associate with the culture of this species [10,44,45,46]. The first case of bivalves infected by *V. alginolyticus* was described in 1965 [47]. They investigated the virulence of *V. alginolyticus* in four different species of bivalve mollusks in different life stages. Later on, Luna-González et al. [48], challenged four bivalve species with *V. alginolyticus* and indicated that scallop species were more susceptible to the pathogen than the other bivalve species tested. In 2005, Gomez-Leon et al. [49], investigated two mortality events happened in 2001 and 2002 in cultured carpet shell clam, *Ruditapes decussatus. V. alginolyticus* was the major causative agent isolated from moribund clam larvae in the first mortality while in the second mortality the same causative agent was isolated alongside with *V. splendidus* Biovar II, causing mortalities up to 62% and 73% of the clam population, respectively. Further, *V. splendidus* strains are mainly linked with mortalities in mussel larvae, although there are many described cases causing mortalities in clams, scallops and in oysters [50,51,52,53,54,55]. *V. tubiashii* was isolated, for the first time, causing mortalities in juvenile clams (*M. mercenariae*) and oyster (*C. virginica*) larvae, in hatcheries located in North America but its first official description was in 1984 [56]. More recently, Prado et al. [57], isolated a new strain of *V. tubiashii,* from flat oyster (*Ostrea edulis*) and Manila clam *(Ruditapes philippinarum*) cultures during an outbreak of the disease in two shellfish hatcheries. *Vibrio tapetis* was responsible for the mass mortalities caused in juveniles and adults in the spring and summer of 1987, reported in the northwest culture beds of France [58]. The disease caused by *V. tapetis*, was named BRD (Brown-Ring-Disease) because of the characteristic brown deposit in the inner part of the valves and affects mostly cultured clams, but has lower impact on wild populations [59]. Brown ring disease is temperature dependent and can cause high mortalities in *R. philippinarum*, but not in *R. decussatus* which has proved to be resistant to BRD [60,61]. *V. crassostreae* can be found in diseased animals and can be characterized pathogenic for pacific oysters, only with the acquisition of a specific plasmid, which is essentially for killing [62]. *Vibrio corallilyticus* was initially described as a coral pathogen and was associated as bivalve pathogen infecting oyster larvae (*Crassostrea virginica* and *Crassostrea gigas*) after challenging oysters in bacterial strains of this *Vibrio* spp. [63,64]. In addition, *M. galloprovinciallis* challenged with *V. corrallilyticus* strains were unable to activate immune responses, although it is considered to be resistant to *Vibrio* infections [65]. Kim et al. [66], investigated the mortality event occurred in a Pacific oyster hatchery and described *V. corallilyticus* as the main etiological agent of this event. *V. mediterranei* was isolated for the first time from plankton, sediments and seawater in two coastal areas in south Valencia, Spain [67]. It is considered to be an emerging pathogen, with a world-wide distribution, having infected corals, cultivated seaweeds and the endangered bivalve *P. nobilis* causing mortalities alongside with other pathogens [38,68,69].It should be emphasized that not all *Vibrio* species act as pathogens for bivalves. Apart from the species considered to be pathogens for the bivalves, there are *Vibrio* spp. that have been detected in the bivalve microbiota in diseased or healthy animals. *Vibrio species*, not associated with mortalities in bivalves, have been detected in bivalve microbiota and they have been initially described living in seawater, in marine crustaceans, in sponges, in corals, in sea horses, in eels and in fish [42,70,71,72,73]. Interestingly, some *Vibrio* species are also of public health importance. Particularly *V. vulnificus*, *V. parahaemolyticus* and *V. cholerae*, are considered to be hazardous in bivalve seafood [3]. In our study, *Vibrio* spp. was identified in all examined specimens (Figure 3). Although, it was not applicable to be unambiguously identified at species level, taking into consideration that *Vibrio* sp. 3 exhibited 99.58% similarity with *V. mediterranei,* the possibility of representing a pathogen for *P. nobilis* contributing in mortalities cannot be ruled out. Additionally, despite forbidden by the local authorities for conservation purposes [21], fishing and consumption of raw fan mussels still occurs (personal communication) and in such a case detected *Vibrio* sp. may be harmful for public health.

Concerning the rest identified bacterial genera, *Photobacterium* species, is the genus expected the most, that can be commonly found in the marine environment and in the intestinal content of marine animals [19]. They constitute one of the oldest genera in the Vibrionaceae and they have been detected in high abundance in the microbiota of *Vibrio*-infected *C. gigas* [11]. Although, most *Photobacterium* spp. have not been described as pathogenic for marine bivalves, isolated from marine bivalves and crustaceans and are common inhabitants of marine environment, subspecies of *Photobacterium damselae* are pathogenic for aquatic animals, especially for fish [74,75,76]. In a recent research, Eggermont et al. [77], isolated bacteria belonging to the family Vibrionaceae from wild blue mussel (*Mytilus edulis*) and assessed their impacts on mussel larviculture. They concluded that most isolates belonged to the genera *Vibrio* and *Photobacterium* and also that *Photobacterium species* with the highest virulence factor in blue mussel was *Photobacterium sanguinicancri. P. sanguinicancri* has been isolated from marine animals previously alongside with other Vibrionaceae [78]. Interestingly, the OTU identified in this research, which is related with *Photobacterium* genera presents similarities 100% with *P. sanguinicancri* as well as with other *Photobacterium* spp. which have been characterized for their symbiotic traits in bivalves [79].

Among the remaining bacterial genera *Aliivibrio, Pseudoalteromonas* and *Psychrilyobacter* constitute probably the most important ones that further discussed.

The genus *Aliivibrio* was first established by Urbanczyk et al. [80], to reclassify *Vibrio* spp. and specifically reconstructed the taxonomy and nomenclature of *V. fischeri*, *V. logei*, *V. salmonicida* and *V. wodanis,* which based on 16s rRNA gene sequencing and after of concatenation of several genes (recA, rpoA, pyrH, gyrB) were renamed as *Aliivibrio* spp. The genus *Aliivibrio* includes species with both symbiotic and pathogenic traits. *A. fischeri* has a worldwide distribution and usually establishes positive relationships with aquatic animals such as cephalopods and fish [81]. *A. logei* has been reported to have symbiotic effects on squids in the genus *Sepiola,* although it is characterized as pathogen of the eels [82,83]. *A. salmonicida* is of great interest on account of its ability to infect fish in low temperature and cause mortality in various marine organisms including bivalves [76,80,84]. OTU identified and related with *Aliivibrio* genus in this research have similarities 100% with all the aforementioned *Aliivibrio* species and therefore ranked as a potential collaborator to rest of the pathogens caused the mortalities in *P. nobilis*.

The genus *Pseudoalteromonas* hosts 41 bacterial species, among which 16 are antimicrobial metabolite producers [85]. Antibacterial activities of *Pseudoalteromonas* spp. utilized in shellfish larviculture as probiotics and achieved higher growth rate, survival rate and higher resistances against infections with *Vibrio* spp. [86,87,88]. Although there are many reports of benefits from the utilization of *Pseudoalteromonas* spp. in shellfish culture, there are also reports of opportunistic behavior of this bacterial genus in mortalities caused by *Vibrio* spp. [89]. We can conclude the dual nature of this genera that act beneficially under the favorite physiological condition of the host and turns into a pathogen when they find the opportunity in a stressful condition.

Similarly, *Psychrilyobacter* spp. was detected in all specimens examined, let alone in high relative abundance, except from the sample originated from the island of Limnos. To our knowledge, there is no report of mortality or even pathogenesis of this genus in bivalves. On the other hand, this specific genus has been detected in the microbiome of bivalves in many cases indicating a beneficial association in disease resistance of the Pacific oysters, *C. gigas* because of the detection of this genus in oysters that were not infected from OsHV-1 virus [90,91,92,93].

Mycoplasmas are widespread in nature and are characterized for their parasite behavior in humans, mammals, reptiles, fish arthropods and plants [94]. Bacteria belonging to the genus *Mycoplasma* infect a wide range of bivalves and have been reported generally as potential bivalve pathogens [59]. First report of this bacterial genus in marine bivalves was by Harshbarger et al. [95], whereas, later, according to Azevedo [96], an unusual branchial infection by a mycoplasma-like microorganism in *C. edule* was reported and associated with high mortalities of this bivalve in Portugal in 1991 and 1992. According to the same study a high prevalence rate (65–70%) of mycoplasma-like microorganisms occurred in gaping cockles and lower abundance in living cockles. He concluded that infection with *Mycoplasma* sp. and high temperatures were the etiological agents of the mortalities. Since then, to our knowledge, no other mortalities have been attributed to *Mycoplasma* spp. as the primary causative agent. King et al. [97], reported the presence of *Mycoplasma* spp. in the stomach microbiome of oysters while in the gut the abundance of the genus wasn’t interestingly high. Lokmer et al. [98], similarly detected high abundance of Mollicutes class, especially bacteria of the genus *Mycoplasma* in the gut of oysters after transplantation of the species in a new environment. There are many other studies investigating the microbiome which have detected the genus *Mycoplasma* as symbiotic bacteria in bivalves [99], while *Mycoplasma* spp. detected in mortalities of *C. gallina* alongside with other genera [19].

In the *P. nobilis* specimens analyzed, relative abundance of Bacteria using a metagenomic approach, demonstrated 14 different OTUs (Average Relative abundance at least 1%) from the examined specimens. Except for *Psychrilyobacter* spp., which according to the literature has not been assigned as a causative agent or as opportunistic genus of bacteria, all other OTUs detected can be potential pathogen for the species. According to our results, four OTUs were detected in relation with *Vibrio* spp. in all specimens in various quantities, which mostly play the primary role in bacterial infections, in agreement with the several cases of mortality events [44]. OTUs with the criteria mentioned above, related with *Mycoplasma* spp. were detected in all samples, with the sample originated from Limnos island showing the highest relative abundance, adding all the OTUs relating with *Mycoplasma* spp. Kal12 was the only sample resulting OTUs in relation with *Mycoplasma* spp. with average relative abundance slightly reaching 0.5%, that was therefore excluded from the generated graphs. It’s worth noting that Lim01 which had the greatest abundance in *Mycoplasma* spp. and the second highest in *Vibrio* spp. exhibited the lowest total bacterial diversity. OTUs in relation with *Photobacterium* spp. were detected in almost all samples except from Ther01 and Lim01 with average abundance degree 1.91%. These results are in line with Milan et al. [19], concerning the detection of *Photobacterium* spp. in mortality events in different bivalve species examined. On the other hand, no great change in abundance of this genus reported in mortality of oysters [20]. OTUs related with *Pseudoalteromonas* spp., were detected in the same specimens as *Photobacterium* spp., in accordance to Li et al. [18], in hemolymph of mussels infected by *V. cyclitrophicus. Pseudoalteromonas* spp. were also present but in lower abundance in the control samples without *V. cyclitrophicus.* Finally, *Aliivibrio* spp. had the lowest average relative abundance in the samples.

Based on our analyses and in line with previous data [34], all specimens, in all status collected, exhibited high abundances of potential pathogens. Bacteria of the genus *Vibrio* were present in all three areas examined showing no correlation with the habitat change. Alongside with the *Vibrio* spp., most of the potential pathogenic bacteria species followed the same pattern and were detected in all three habitats. Additionally, there were no differences in *Vibrio* spp. abundances between the samples collected in the winter (Kal10, Kal12) and the samples collected during warmer months (Ther01, Ther02, Ther03 and Lim01), leading to the conclusion of the synergistical effect of those bacterial species in mortality events. Interestingly, the only bacterial species with high correlation to the habitat status were bacteria belonging to the genus *Mycoplasma* spp. that appeared to play a key role in the microbiota of *P. nobilis* originated from Limnos island, decreasing also the gut microbial diversity. Our results are in accordance with bacteriological study in *P. nobilis* implemented in three separate locations located in Eastern Adriatic coast [100], also demonstrating the presence of *V. splendidus* clade bacteria hosting *P. nobilis*.

However, to our knowledge, this is the first study approaching *P. nobilis* microbiome using a 16s-rRNA Metagenomic tool. Overall, in the present study many potential pathogens were determined, which, may be considered to act as potential pathogens or may have a secondary role in mortality events.Although the use of 16s-rRNA may have some limitations in the distinction of bacteria at species level and despite the fact that the examination of 6 gut microbiomes is only relatively sufficient for a general conclusion, our study demonstrated that the microbial diversity in diseased *P. nobilis* specimens contains specific genera of bacteria of high pathogenic importance. These bacterial genera occur in all geographical areas where mortalities took place, having similar abundance levels. Excluding the unique individual originated from the last collapsed population of Limnos island, the remaining investigated populations demonstrate the same bacteria genera with slight differences in abundances, although all of these populations have collapsed. We therefore assume that mortalities may be due to several bacterial genera and not only on account of the genera already reported. Devastating mortality events in *P. nobilis* populations are threatening fan mussel towards extinction. At the beginning, the *H. pinnae* was detected and the *Mycobacterium* sp., almost at the same time, were targeted as possible causes for the mortalities in the Mediterranean Sea. Mycobacteriosis is a chronic disease, probably influenced by the climate change, infecting both marine and terrestrial animal species with underestimated impacts on infection of aquatic animals [101]. Mycobacterial infections in marine bivalves are in scarce, although in 2019 and 2020 two cases of infections were reported, playing a significant role in mortality events of the endangered bivalve *P. nobilis* in the Mediterranean Sea [34,37]. Previously, Grimm et al. [102], reported the presence of a *Mycobacterium* sp. in the Atlantic sea scallop *Placopecten magellanicus,* as causative agent of orange nodular lesions. However, following the recent results of the studies concerning the mortalities, new pathogens have arisen alongside and seemed to play a key role in mortalities of *P. nobilis*. Prado et al. [38], exhibited a new pathogen in their effort to rescue and relocate *P. nobilis* individuals in close water systems. Peak of mortalities occurred when the temperatures reached above 25 °C and the causative agent was *V. mediterranei*, a newly unknown pathogen until that time in fan mussel, *P. nobilis*. In addition, Carella et al. [39], illustrated all the aforementioned pathogens plus a *Perkinsus* sp. in two specimens examined which indicated one more potential threat for the fan mussel. In order to unravel the puzzling situation of the collapse of *P. nobilis* populations, the occurrence of *Rhodococcus erythropolis* arose as a new bacteria species having a potential role in mortality events [103]. Fan mussel populations are collapsing one after the other in Mediterranean Sea. Recently, an epidemiological study showed the rapid progression even in the last safe places in Croatia [104]. In Greece, the population of the species suffered enormous damage, limiting them in few areas but with the presence of *Mycobacterium* sp. and *H. pinnae*. These last population hot spots are suffering from mortalities mostly on warmer months. In conclusion, according to our results and in line the aforementioned recent studies, probably *P. nobilis* mortalities are not caused solely by the infection with *H. pinnae* and *Mycobacterium* sp. as previously assumed, but also, other pathogenic bacterial genera may act synergistically with these parasites, resulting in the observed mortalities.

## Figures and Tables

**Figure 1 pathogens-09-01002-f001:**
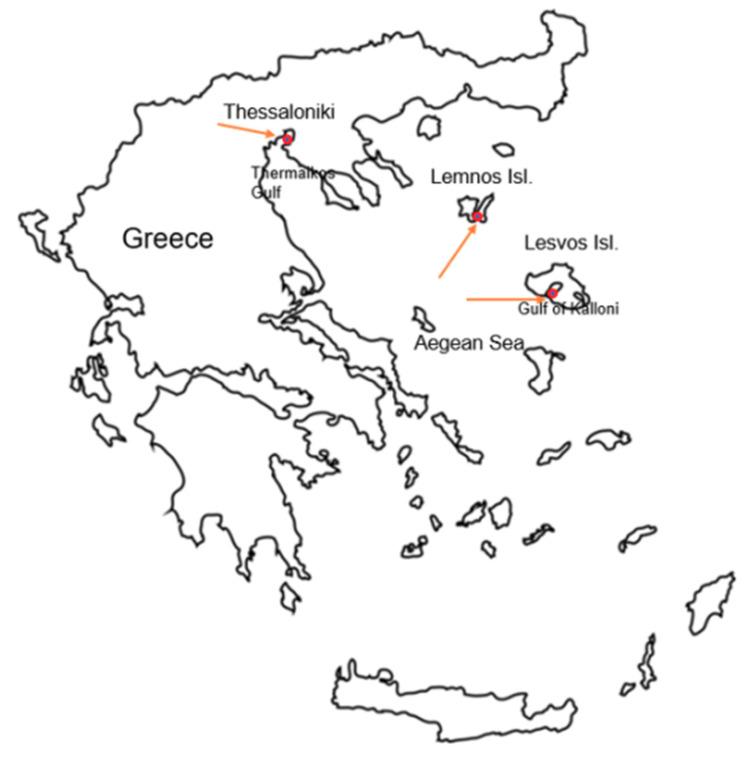
Sampling areas of *Pinna nobilis* specimens, collected for the needs of the microbial investigation [34].

**Figure 2 pathogens-09-01002-f002:**
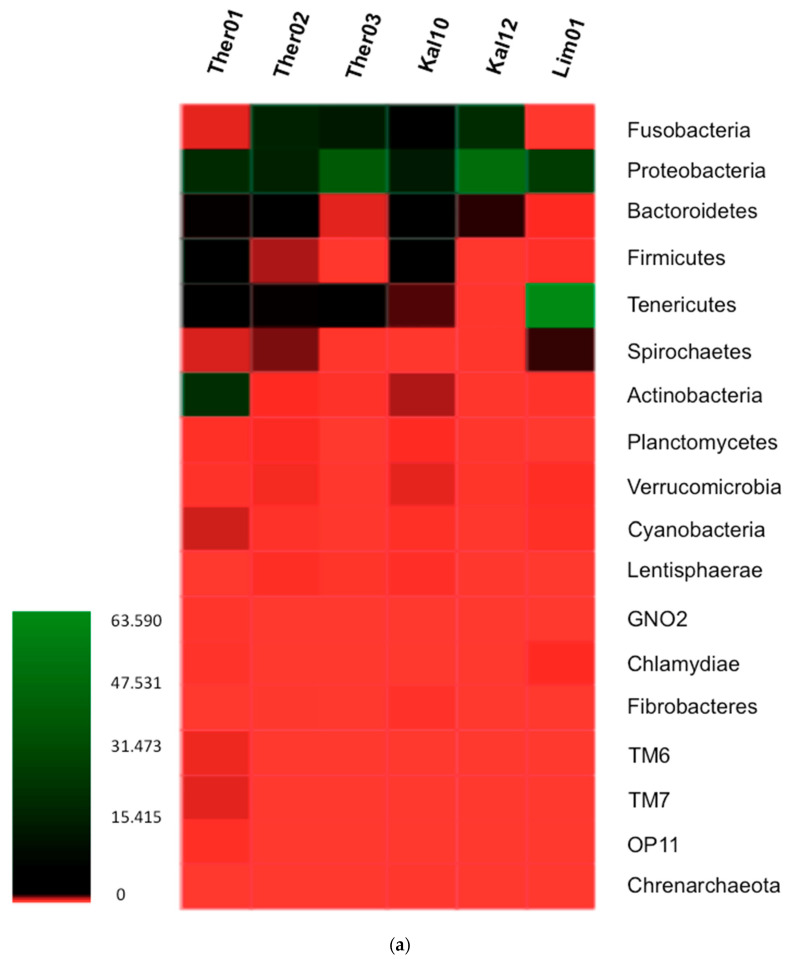
(**a**) Relative abundance of bacterial communities of infected *P. nobilis* digestive gland at phylum level, including the archaea phylum Crenarchaeota, presented in a heatmap. (**b**) Relative abundance of bacterial phyla in infected *P. nobilis* specimens presented in bar graph.

**Figure 3 pathogens-09-01002-f003:**
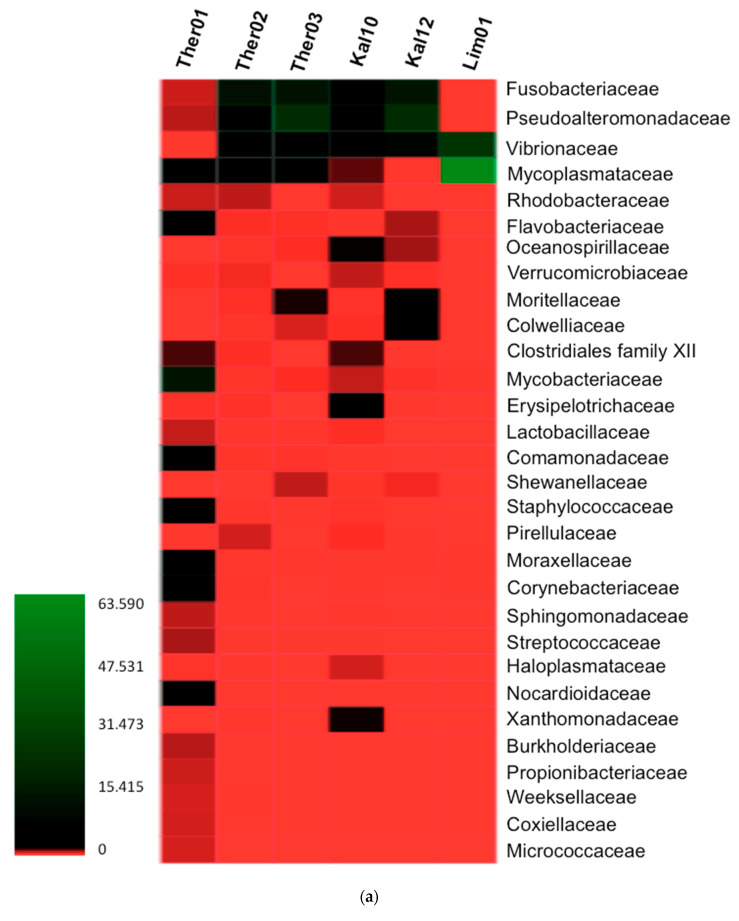
(**a**) Relative abundance of bacterial communities of infected *P. nobilis* digestive gland at family level, represented in a heatmap. (**b**) Relative abundance of bacterial families in infected *P. nobilis* specimens.

**Figure 4 pathogens-09-01002-f004:**
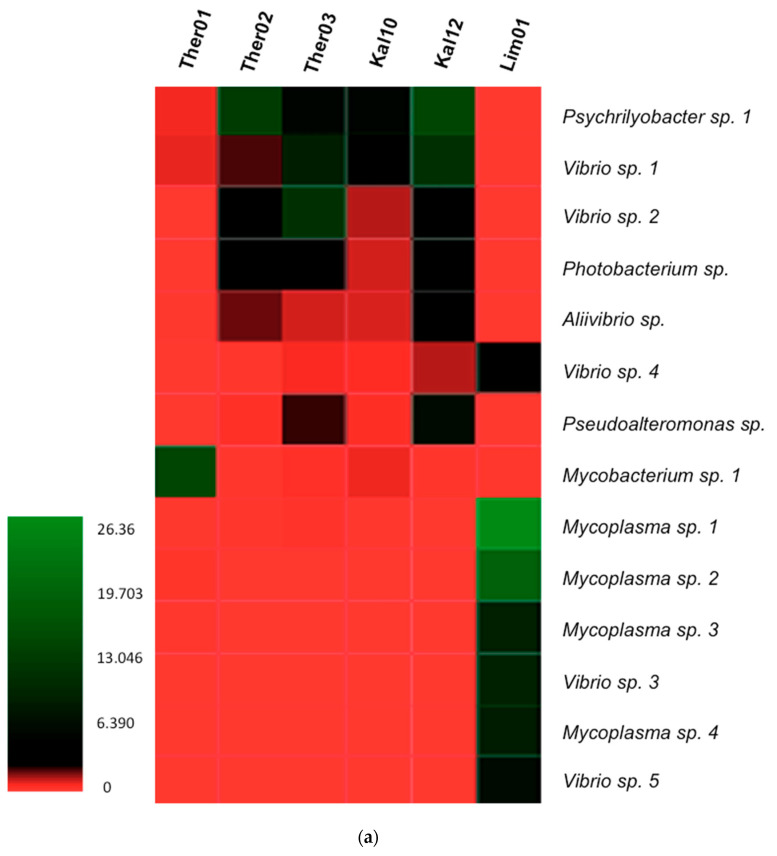
(**a**) Bacterial diversity of infected *P. nobilis* specimens in genus level, documented in a heatmap. (**b**) Relative abundance of bacterial genera in infected *P. nobilis* specimens.

**Table 1 pathogens-09-01002-t001:** Identities of the samples originated from Aegean Sea. [34].

Specimen id	*H. pinnae*	*Mycobacterium* spp.
Kal10		+
Kal12	+	+
Lim01	+	+
Ther01		+
Ther02	+	+
Ther03	+	+

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
