# Peer review of "Gut Symbiotic Microbial Communities in the IUCN Critically Endangered Pinna nobilis Suffering from Mass Mortalities, Revealed by 16S rRNA Amplicon NGS"

_pathogens, 2020, doi:10.3390/pathogens9121002_

Round 1
Reviewer 1 Report
Dear authors,
I found your work quite interresting in the context of mass mortality events of Pinna nobilis.
I only have few minor comments:
- I guess you should justify that the use of 6 gut microbiome is relevant enough for a general conclusion.
- Line 147: replace "both of the pathogens" by H. pinnae and Mycobaterium sp. as in the current form you only refer to Mycobacterium sp.
- Figure 1 : better indicate the sampling areas as the red dot appear too smalls.
- Figure 2 : replace (alpha) by (a)
- Figures of heatmap : please make better legible figures.
Author Response
We would like to thank the first reviewer for appraising the interest of the study. In the following lines, there is a detailed response for each of his/her comments
Comment 1: I guess you should justify that the use of 6 gut microbiome is relevant enough for a general conclusion.
Response: The recommendation of the reviewer was taken into consideration and the fact that "the examination of 6 gut microbiomes is only relatively sufficient for a general conclusion" was clarified and added in the Discussion (lines 446-447 in the revised manuscript)
Comment 2: Line 147: replace "both of the pathogens" by H. pinnae and Mycobaterium sp. as in the current form you only refer to Mycobacterium sp.
Response:Corrected according the suggestions of the reviewer (Line 150 in revised manuscript)
Comment 3: Figure 1: better indicate the sampling areas as the red dot appear too smalls.
Response:Figure 1 was recreated and specific arrows were added in order to indicate the sampling spots.
Comment 4: Figure 2 : replace (alpha) by (a)
Response: Corrected according to reviewer’s comments (Line 197 in revised manuscript).
Comment 5: Figures of heatmap: please make better legible figures.
Response: The heatmaps were replaced by more legible ones, of better quality
Reviewer 2 Report
The manuscript is original and represent a very interesting piece of work related to the mass mortality of Pinna nobilis.
I recommend the publication of the manuscript after few small improvements:
1. In the text scientific names are written with the abbreviation of the genus (e.g. P. nobilis) and with full genus (e.g. Pinna nobilis) with a no logic way. Please standardize with the whole text.
2. The paper is focused on etiological agents of Pinna nobilis which are causing tha mass mortality. However, the host species Pinna nobilis is the real protagonist of the event. I would like to read an improvement of the background on P. nobilis in the introduction, at least for what its past conservation status is concerned.
For instance, in line 112 the species has been introduced and after two line author wrote that it has been recently included in the IUCN Red List of threatened species. However the species during years went through a threat since 1980s as consequence of anthropogenic activities (see e.g. Rabaoui et al., 2010 and Vázquez‐Luis et al., 2015).
As a consequence of these threats, P. nobilis was included in a
protection regime (Barcelona Convention and EU Habitats Directive) which led the species to to a recovery of its populations, as already shown also by genetic analyses (Sanna et al., 2003, 2004). Unfortunately, recently ........
References
Rabaoui, L.; Tlig‐Zouari, S.; Katsanevakis, S.; Hassine, O.K.B. Modelling population density of Pinna nobilis (Bivalvia) on the eastern and southeastern coast of Tunisia. J. Molluscan Stud. 2010, 76, 340–347, doi:10.1093/mollus/eyq023.
Vázquez‐Luis, M.; Borg, J.A.; Morell, C.; Banach‐Esteve, G.; Deudero, S. Influence of boat anchoring on Pinna nobilis: A field experiment using mimic units. Mar. Freshw. Res. 2015, 66, 786–794, doi:10.1071/mf14285.
Sanna, D.; Cossu, P.; Dedola, G.L.; Scarpa, F.; Maltagliati, F.; Castelli, A.; Franzoi, P.; Lai, T.; Cristo, B.; Curini‐Galletti, M.; et al. Mitochondrial DNA Reveals Genetic Structuring of Pinna nobilis across the Mediterranean Sea. PLoS ONE 2013, 8, e67372, doi:10.1371/journal.pone.0067372.
Sanna, D.; Dedola, G.; Scarpa, F.; Lai, T.; Cossu, P.; Curini‐Galletti, M.; Francalacci, P.; Casu, M. New mitochondrial and nuclear primers for the Mediterranean marine bivalve Pinna nobilis. Mediterr. Mar. Sci. 2014, 15, 416, doi:10.12681/mms.459.
Author Response
We are grateful to the reviewer for his/her nice opinion comments regarding the originality and the interest of our manuscript. All improvements proposed by the reviewer were carried out and are detailed in the following lines:
Comment 1: In the text scientific names are written with the abbreviation of the genus (e.g. P. nobilis) and with full genus (e.g. Pinna nobilis) with a no logic way. Please standardize with the whole text.
Response: Pinna nobiliswas replaced by P. nobilisin the entire manuscript, apart from the first time mentioned, as well as from the titles of the sections where referred.
Comment 2: The paper is focused on etiological agents of Pinna nobilis which are causing tha mass mortality. However, the host species Pinna nobilis is the real protagonist of the event. I would like to read an improvement of the background on P. nobilis in the introduction, at least for what its past conservation status is concerned.
For instance, in line 112 the species has been introduced and after two line author wrote that it has been recently included in the IUCN Red List of threatened species. However the species during years went through a threat since 1980s as consequence of anthropogenic activities (see e.g. Rabaoui et al., 2010 and Vázquez‐Luis et al., 2015).
As a consequence of these threats, P. nobilis was included in a
protection regime (Barcelona Convention and EU Habitats Directive) which led the species to to a recovery of its populations, as already shown also by genetic analyses (Sanna et al., 2003, 2004). Unfortunately, recently ........
References
Rabaoui, L.; Tlig‐Zouari, S.; Katsanevakis, S.; Hassine, O.K.B. Modelling population density of Pinna nobilis (Bivalvia) on the eastern and southeastern coast of Tunisia. J. Molluscan Stud. 2010, 76, 340–347, doi:10.1093/mollus/eyq023.
Vázquez‐Luis, M.; Borg, J.A.; Morell, C.; Banach‐Esteve, G.; Deudero, S. Influence of boat anchoring on Pinna nobilis: A field experiment using mimic units. Mar. Freshw. Res. 2015, 66, 786–794, doi:10.1071/mf14285.
Sanna, D.; Cossu, P.; Dedola, G.L.; Scarpa, F.; Maltagliati, F.; Castelli, A.; Franzoi, P.; Lai, T.; Cristo, B.; Curini‐Galletti, M.; et al. Mitochondrial DNA Reveals Genetic Structuring of Pinna nobilis across the Mediterranean Sea. PLoS ONE 2013, 8, e67372, doi:10.1371/journal.pone.0067372.
Sanna, D.; Dedola, G.; Scarpa, F.; Lai, T.; Cossu, P.; Curini‐Galletti, M.; Francalacci, P.; Casu, M. New mitochondrial and nuclear primers for the Mediterranean marine bivalve Pinna nobilis. Mediterr. Mar. Sci. 2014, 15, 416, doi:10.12681/mms.459.
Response: As suggested by the reviewer, information about the declining population status was added in Lines 114-123 in the revised manuscript. We have also added the aforementioned studies in the reference list in order to provide a more detailed description of the conservation status of Pinna nobilis.
Reviewer 3 Report
I found the manuscript really interesting and well written. It provides very important results to investigate the pathogenesis of MME in Pinna nobilis. After several reads, I have not found minor revisions. My compliments to the authors.
Author Response
We would like to express our gratefulness to the reviewer for his/her opinion. We hope that the article would be an important contribution to the pathogenesis of MME in Pinna nobilis.
This manuscript is a resubmission of an earlier submission. The following is a list of the peer review reports and author responses from that submission.
Round 1
Reviewer 1 Report
The paper of Lattos et al deals with the microbial community explored by 16S rRNA next generation sequencing (NGS) of individuals of the endangered species Pinna nobilis involved in the mass mortality events (MME)s in Greece, following a recent publication on the topic.
NGS of the 16S rRNA encoding region is a powerful instrument, a useful molecular target since it is present in all bacteria, recently used to characterize the microbial population in disease and healthy condition, in plant, animals and humans. Even if the paper of Lattos et al can have some merit in trying to unravel the big puzzle around the death of this threaten species, probably going to extinction, the paper can’t be accepted for publication. It’s in my opinion incomplete in many aspects.
I don’t think the study as conducted, with only sick individuals for NGS, is correct. First big issue is the lack of healthy animals in the analysis. Unacceptable. How do you define a threshold if it is the first time the authors make this type of study in a species where there are no data in regard at all?
Moreover, the authors report the different bacterial species identified, but this method does not allow to identify bacteria to the species level due to high sequence similarities between some species. Authors should, after this first approach, go to species level and eventual pathogen pathogenicity/virulence. What is the sense of define the presence of Vibrio spp. If we don’t know what species? The same thing about Mycoplasma, the authors report as highly represented. They should define what species, it’s potential virulence etc..
Moreover, what is the sense of this analysis if they can’t correlate them with some lesions and morphology. If we describe a disease condition only with the use of big data, we should also recognize its limitation.
Reviewer 2 Report
The authors choice of topic for manuscript ref.# 949944 is timely as the host species under study is suffering from region-wide mortality events of questionable aetiology. The work is not ready for publication for several reasons summarized below.
1) The English language is in need of professional revision. There are many awkward constructs that leave room for doubt as to their intended meaning. There are small errors throughout as well such as spaces missing between the end of one sentence and the beginning of the next. Examples of poor use of English:
Line 20 "toward the causes" not "on the causes"
Line 40: "arose" rather than "came up"
Line 73: should be - "Concerning the microbiome of bivalves..."
Line 80: "through" rather than "though"
Line 109: What is the "inhibitor factor"? Antibiotics?
2) Given the Mediterranean-wide distribution of the host species and the variety of islands in Greece, it would be useful to readers to have a map illustrating the sampling sites to give a more clear picture as to the samples that are referenced.
3) Heat Maps are a poor choice for illustrating this type of data. The color scale lacks intuitive quantitative information (given the scale and color scheme) and this type of information is more normally represented in pie graphs or bar graphs. Figures need complete revision.
4) Also it is non-sense, pure and simple, to talk about Mycobacteria species and Vibrio species when the data obtained only provides information at the level of genus. For Vibrio especially, the 16S rDNA is a poor marker for species delineation, anyway. All of this text needs to be rethought and re-written. The author might consider as well the recent work on a species-specific PCR for Vibrio mediterranei (found in Pinna nobolis), as this could provide additional data about specific pathogenic species from Pinna nobilis (Andree, et al., 2020, Journal of Applied Microbiology).
The authors are also strongly advised to read,
Razin S, Yogev D, Naot Y. Molecular biology and pathogenicity of mycoplasmas. Microbiol Mol Biol Rev. 1998;62(4):1094-1156.
... to get a better appreciation for Mycoplasma and Molicutes generally. Due to reductionist evolution in this group of bacteria there are many that are obligate commensals or symbionts and not necessarily pathogens. Identification of this bacterial group in the intestine is quite common among nearly all extant metazoan life.